

# Bombardier beetles repel invasive bullfrogs

Shinji Sugiura and Tomoki Date

Graduate School of Agricultural Science, Kobe University, Kobe, Hyogo, Japan

## ABSTRACT

Invasive non-native predators negatively affect native species; however, some native species can survive the predation pressures of invasive species by using pre-existing antipredator strategies or evolving defenses against invasive predators. The American bullfrog *Lithobates catesbeianus* (Anura: Ranidae) has been intentionally introduced to many countries and regions, and has impacted native animals through direct predation. Bombardier beetles (Coleoptera: Carabidae: Brachininae: Brachinini) discharge chemicals at a temperature of approximately 100 °C from the tip of the abdomen when they are attacked by predators. This "bombing" can successfully repel predators. However, adults of a native bombardier beetle *Pheropsophus* (*Stenaptinus*) *occipitalis jessoensis* have been reportedly found in the gut contents of the introduced bullfrog *L. catesbeianus* in Japan. These records suggest that the invasive bullfrog *L. catesbeianus* attacks the native bombardier beetle *P. occipitalis jessoensis* under field conditions in Japan; however, the effectiveness of the bombing defense against invasive bullfrogs is unclear. To test the effectiveness of the bombing defense against bullfrogs, we investigated the behavioral responses of *L. catesbeianus* juveniles to *P. occipitalis jessoensis* adults under laboratory conditions. Contrary to previous gut content results, almost all the bullfrogs (96.3%) rejected bombardier beetles before swallowing them; 88.9% rejected the beetles after being bombed, and 7.4% stopped attacking the beetles before being bombed. Only 3.7% successfully swallowed and digested the beetle. All of the beetles collected from non-bullfrog-invaded sites could deter bullfrogs, suggesting that the pre-existing defenses of bombardier beetles played an essential role in repelling bullfrogs. When treated beetles that were unable to discharge hot chemicals were provided, 77.8% of bullfrogs successfully swallowed and digested the treated beetles. These results indicate that bombing is important for the successful defense of *P. occipitalis jessoensis* against invasive bullfrogs. Although invasive bullfrogs have reportedly impacted native insect species, *P. occipitalis jessoensis* has an existing defense mechanism strong enough to repel the invasive predators.

## INTRODUCTION

Invasive non-native species negatively impact native biota (*Doherty et al., 2016*; *Sugiura, 2016*; *David et al., 2017*). In particular, invasive predators affect native communities and ecosystems through cascading effects (*Goldschmidt, Witte & Wanink, 1993*; *O'Dowd, Green & Lake, 2003*; *Kenis et al., 2009*; *David et al., 2017*; *Rogers et al., 2017*; *McGruddy*

Corresponding author
Shinji Sugiura,
ssugiura@people.kobe-u.ac.jp,
sugiura.shinji@gmail.com

*et al., 2021*). Because native prey species do not share a history with invasive predators (*Fritts & Rodda, 1998*; *Strauss, Lau & Carroll, 2006*; *Carthey & Banks, 2014*), many native species suffer predation by invasive species (*Goldschmidt, Witte & Wanink, 1993*; *Doherty et al., 2016*; *Sugiura, 2016*). However, some native species have survived the predation pressures of invasive species by using pre-existing antipredator strategies (*Davis, Epp & Gabor, 2012*; *Carthey & Banks, 2014*) or evolving defenses against invasive predators (*Vermeij, 1982*; *Strauss, Lau & Carroll, 2006*). Pre-existing antipredator defenses can play an important role in repelling invasive predators that have similar ecological traits to native predators (*Carthey & Banks, 2014*; *Melotto et al., 2021*). However, pre-existing defenses have received less attention than the evolution of anti-predator defenses in terms of native species' tolerance to invasive predators (*Strauss, Lau & Carroll, 2006*). Investigating the effectiveness of the pre-existing defenses of native species against invasive predators would enable a better understanding of how to mitigate the impacts of invasive predators on native species.

The American bullfrog *Lithobates catesbeianus* (Shaw) (formerly called *Rana catesbeiana* Shaw; *Lowe et al., 2000*) (Anura: Ranidae) has been intentionally introduced for various purposes to many countries and regions (western North America, South America, East and Southeast Asia, and Western Europe) from eastern North America (*Ficetola et al., 2007*; *Ficetola, Thuiller & Miaud, 2007*; *Giovanelli, Haddad & Alexandrino, 2008*; *Bissattini & Vignoli, 2017*; *Groffen et al., 2019*; *Johovic et al., 2020*). Eggs are laid in still water such as ponds (*Govindarajulu, Price & Anholt, 2006*). The larvae feed on algae, diatoms, cyanobacteria, protists, and tiny invertebrates in water (*Kupferberg, 1997*; *Pryor, 2003*; *Ruibal & Laufer, 2012*). Postmetamorphic juveniles and adults prey on various animals (including aquatic and terrestrial species) in and near water (*Hirai, 2004*; *Govindarajulu, Price & Anholt, 2006*; *Dontchev & Matsui, 2016*; *Flynn, Kreofsky & Sepulveda, 2017*; *Laufer et al., 2021*; *Sarashina & Yoshida, 2021*). Because bullfrog adults commonly reach a size (snout–vent length) of 180–200 mm (*Werner, Wellborn & McPeek, 1995*), they are able to swallow small vertebrates (*e.g.*, fish, mammals, reptiles, and frogs) as well as invertebrates (*Raney & Ingram, 1941*; *Stewart & Sandison, 1972*; *Bruneau & Magnin, 1980*; *Clarkson & DeVos, 1986*; *Govindarajulu, Price & Anholt, 2006*; *Flynn, Kreofsky & Sepulveda, 2017*; *Oda et al., 2019*). Consequently, invasive bullfrogs have impacted native communities in invaded habitats (*Kats & Ferrer, 2003*; *Li et al., 2011*; *Adriaens, Devisscher & Louette, 2013*; *Gobel, Laufer & Cortizas, 2019*). Therefore, *L. catesbeianus* has been listed as one of the 100 "world's worst invaders" (*Lowe et al., 2000*). Many studies have investigated the gut or stomach contents of adult and juvenile bullfrogs in native (*Raney & Ingram, 1941*; *Korschgen & Moyle, 1955*; *Fulk & Whitaker, 1968*; *Stewart & Sandison, 1972*; *Bruneau & Magnin, 1980*; *Werner, Wellborn & McPeek, 1995*) and invaded (*Clarkson & DeVos, 1986*; *Balfour & Morey, 1999*; *Krupa, 2002*; *Hirai, 2004*, *2005*; *Wu et al., 2005*; *Hirai & Inatani, 2008*; *Mori, 2008*; *Silva et al., 2009*; *Barrasso et al., 2009*; *Leivas, Leivas & Moura, 2012*; *Boelter et al., 2012*; *Jancowski & Orchard, 2013*; *Ortíz-Serrato, Ruiz-Campos & Valdez-Villavicencio, 2014*; *Quiroga et al., 2015*; *Liu et al., 2015*; *Dontchev & Matsui, 2016*; *Flynn, Kreofsky & Sepulveda, 2017*; *Vrcibradic et al., 2017*; *Park et al., 2018*; *Bissattini, Buono & Vignoli, 2018*, *2019*; *Oda et al., 2019*; *Matsumoto,*

*Suwabe & Karube, 2020*; *Laufer et al., 2021*; *Nakamura & Tominaga, 2021*) ranges, with the results indicating that introduced bullfrogs frequently attack native animal species in invaded areas. However, few studies have directly observed how invasive bullfrogs can attack and swallow native prey. Investigating the attack behavior of bullfrogs would help to assess which native species suffer from bullfrog predation.

Carabidae is one of the most diverse families in Coleoptera. Carabid beetles have frequently been used as bioindicators (*Rainio & Niemelä, 2003*) and biocontrol agents (*Kromp, 1999*). Carabid adults also exhibit morphological, physiological, chemical, and behavioral defenses against predators (*Giglio et al., 2021*). For example, adult bombardier beetles (Coleoptera: Carabidae: Brachininae: Brachinini) discharge toxic chemicals (*e.g.*, 1,4-benzoquinone and 2-methyl-1,4-benzoquinone) and water (vapor) at a temperature of approximately 100 °C (*i.e.*, bombing) from the tip of abdomen when they are attacked by predators (Video S1; *Aneshansley et al., 1969*; *Kanehisa & Murase, 1977*; *Dean, 1979*; *Kanehisa, 1996*; *Eisner, Eisner & Siegler, 2005*; *Arndt et al., 2015*; *Sugiura, 2018*, *2021*). The hot chemicals can effectively protect the beetles from predators such as arthropods (*Eisner, 1958*; *Eisner & Meinwald, 1966*; *Eisner & Dean, 1976*; *Eisner et al., 2006*; *Sugiura, 2021*), amphibians (*Eisner & Meinwald, 1966*; *Dean, 1980*; *Sugiura & Sato, 2018*; *Sugiura, 2018*), reptiles (*Bonacci et al., 2008*), and birds (*Kojima & Yamamoto, 2020*).

The bombardier beetle *Pheropsophus* (*Stenaptinus*) *occipitalis jessoensis* Morawitz (formerly called *Pheropsophus jessoensis* Morawitz; *Sugiura & Sato, 2018*; *Sugiura, 2018*, *2021*), which is commonly found in grassland, farmland, and forest edge environments in Japan, Korea, China, and Vietnam (*Habu & Sadanaga, 1965*; *Yahiro et al., 1992*; *Ishitani & Yano, 1994*; *Fujisawa, Lee & Ishii, 2012*; *Ohwaki, Kaneko & Ikeda, 2015*; *Fedorenko, 2021*), has been frequently studied to investigate the effectiveness of bombing as an anti-predator defense (*Sugiura, 2018*; *Sugiura & Sato, 2018*; *Kojima & Yamamoto, 2020*; *Sugiura, 2021*). *Pheropsophus occipitalis jessoensis* can successfully deter birds (*Kojima & Yamamoto, 2020*), frogs (*Sugiura, 2018*), and praying mantises (*Sugiura, 2021*). However, adults of the bombardier beetle *Ph. occipitalis jessoensis* have been reportedly found in the stomach contents of field-collected bullfrogs in central Japan; for example, two adult beetles were found in a dead bullfrog (*Mori, 2008*) and an adult beetle was found in a juvenile bullfrog (*Matsumoto, Suwabe & Karube, 2020*). These records suggest that the invasive bullfrog *L. catesbeianus* attacks the native bombardier beetle *Ph. occipitalis jessoensis* under field conditions in Japan, but the bombing defense of *Ph. occipitalis jessoensis* against invasive bullfrogs remains unexplored. To test the effectiveness of the bombing defense against bullfrogs, we investigated how *L. catesbeianus* juveniles respond to *Ph. occipitalis jessoensis* adults under laboratory conditions. In addition, the responses of bullfrogs to *Ph. occipitalis jessoensis* collected from bullfrog-invaded sites were compared with those of beetles collected from non-invaded sites to investigate whether native bombardier beetles that coexist with invasive bullfrogs exhibit a stronger defense than beetles that do not coexist with bullfrogs.

## MATERIALS AND METHODS

### Study species

To investigate how bullfrogs respond to bombardier beetles under laboratory conditions, we used juveniles of the invasive bullfrog *L. catesbeianus* and adults of the bombardier beetle *Ph. occipitalis jessoensis*. We observed *Ph. occipitalis jessoensis* adults, *L. catesbeianus* juveniles, and native pond frogs *Pelophylax nigromaculatus* (Hallowell) (Anura: Ranidae) in the same grassland around a pond in Hyogo, central Japan, on the same date (Fig. 1). Therefore, bullfrog juveniles may frequently encounter adults of *Ph. occipitalis jessoensis* in grassland around ponds, lakes, and paddy fields in Japan.

Fifty-four juvenile bullfrogs (snout–vent length: 42.2–59.6 mm) were collected from grassland around a pond in Hyogo, Japan, between August and October 2021. The bombardier beetle *Ph. occipitalis jessoensis* was frequently found in this sampling site. The snout–vent length and body weight of each bullfrog were measured to the nearest 0.01 mm and 0.1 mg, using an electronic slide caliper (CD-S15C; Mitutoyo, Kanagawa, Japan) and an electronic balance (CPA64; Sartorius Japan K.K., Tokyo, Japan), respectively. Juvenile bullfrogs were maintained separately in small plastic cages (120 × 85 × 130 mm, length × width × height) in the laboratory at 25 °C (cf. *Sugiura, 2018*, *2020b*). Live mealworms (larvae of *Tenebrio molitor* Linnaeus (Coleoptera: Tenebrionidae)) were provided as food (cf. *Sugiura, 2018*, *2020b*). Bullfrogs were starved for 24 h before the experiments to standardize their hunger level (cf. *Sugiura, 2018*, *2020b*). Individual bullfrogs were not used repeatedly (cf. *Sugiura, 2018*, *2020b*). Introduced bullfrogs have been designated as an "invasive alien species" in Japan. Therefore, transportation, laboratory keeping, and behavioral experiments of bullfrogs were performed with permission from the Kinki Regional Environmental Office of the Ministry of the Environment, Government of Japan (Number: 20000085).

Fifty-four adults of the bombardier beetle *Ph. occipitalis jessoensis* (body length: 15.2–20.2 mm) were collected from grasslands and farmlands in Hyogo (three sites), Shiga (one site), Kyoto (one site), and Shimane (one site), central Japan (Fig. S1), in July–September 2020 and May–October 2021; 39 and 15 beetles were collected from bullfrog-invaded sites (three sites in Hyogo and one in Shimane) and non-invaded sites (one site in Kyoto and one in Shiga), respectively (Fig. S1). All adult beetles displayed bombing when manually caught by our researchers under field conditions. Body length and body weight of each beetle were also measured. Bombardier beetles were maintained separately in small plastic cases (diameter: 85 mm; height: 25 mm) in the laboratory at 25 °C (cf. *Sugiura, 2018*; *Sugiura & Sato, 2018*; *Sugiura, 2021*). Dead larvae of *Spodoptera litura* (Fabricius) (Lepidoptera: Noctuidae) were provided as food (cf. *Sugiura, 2018*; *Sugiura & Sato, 2018*; *Sugiura, 2021*). Individual beetles were not used repeatedly (cf. *Sugiura, 2018*; *Sugiura & Sato, 2018*; *Sugiura, 2021*).

### Experiments

Following the method of *Sugiura (2018*, *2020b)*, we investigated how bullfrogs can attack bombardier beetles in our laboratory (25 °C) between September and November 2021.

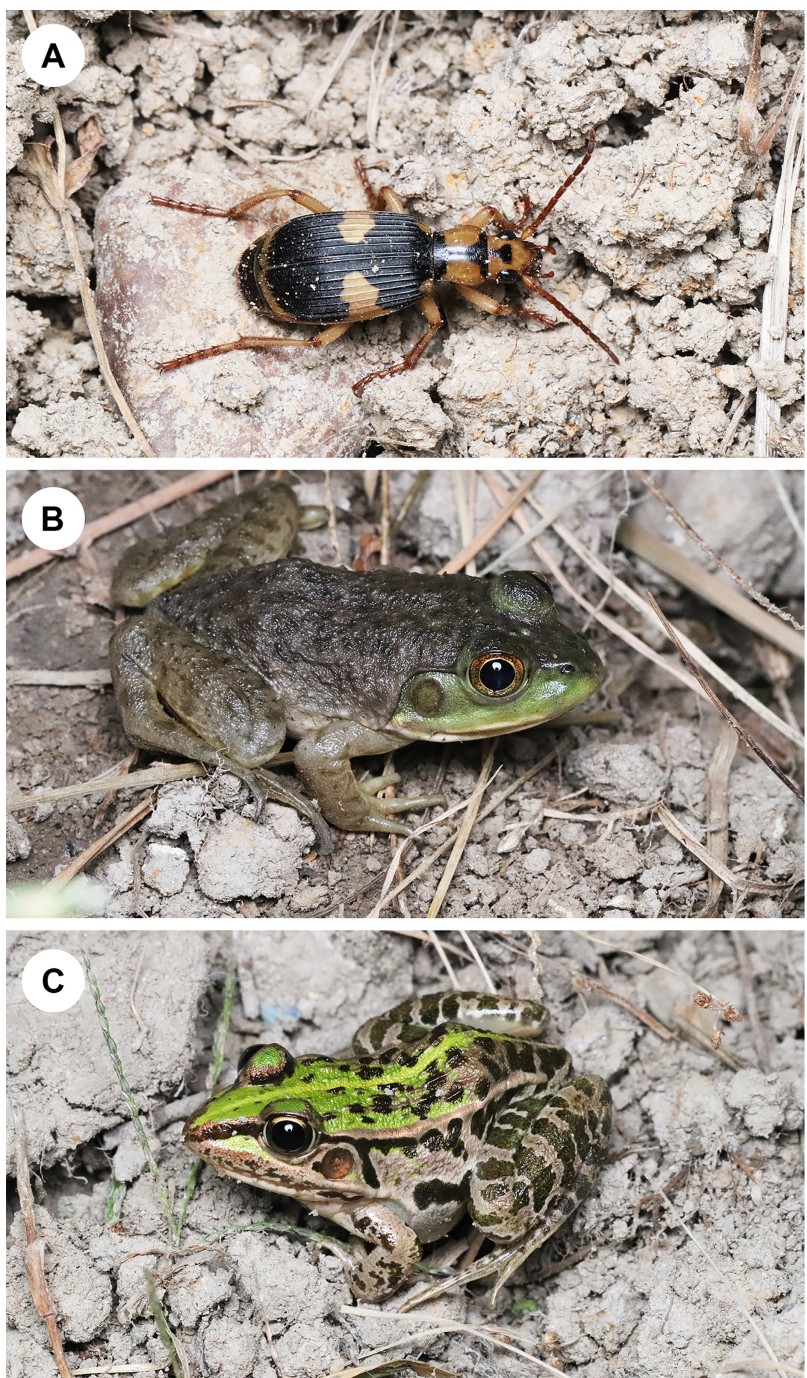

**Figure 1** **A bombardier beetle, an invasive bullfrog, and a native frog.** (A) An adult bombardier beetle *Pheropsophus occipitalis jessoensis*. (B) A juvenile bullfrog *Lithobates catesbeianus*. (C) An adult pond frog *Pelophylax nigromaculatus*. These photographs were taken at the same site and microhabitat on the same date. Photo credit: Shinji Sugiura.

Bullfrogs that ate mealworms >1 day before experiments were used. We were unable to sex either bullfrogs or beetles due to their age and morphology, respectively. First, we placed a bullfrog in a plastic cage (120 × 85 × 130 mm, length × width × height). Then, we placed an

adult beetle in the cage. We recorded the behavior of the bullfrog and the beetle using digital cameras (iPhone 12 Pro Max; Apple Inc., Cupertino, CA, USA; Handycam HDR-CX630V; Sony, Tokyo, Japan). When a bullfrog rejected a beetle, we played back the footage of the recorded behavior to investigate whether the rejection was due to bombing. When bombing sounds were heard or ejected vapor was seen, we considered that bombing forced the bullfrog to reject the beetle. When a bullfrog swallowed a beetle, we investigated whether it vomited the beetle (cf. *Sugiura, 2018*; *Sugiura & Sato, 2018*). Bullfrogs that did not vomit were considered to have digested the beetle. When a bullfrog did not swallow a beetle, we provided a mealworm as a palatable prey to the bullfrog several minutes after beetle rejection to determine whether this rejection was due to satiation (cf. *Sugiura, 2018*, *2020b*). In total, 27 bullfrogs and 27 beetles were used in this experiment.

To test the role of hot chemical ejection by bombardier beetles in deterring bullfrogs, we provided bullfrogs with treated *Ph. occipitalis jessoensis* that were unable to eject hot chemicals (thereafter, treated beetles; cf. *Sugiura & Sato, 2018*; *Sugiura, 2021*). Following the method of *Sugiura & Sato (2018)* and *Sugiura (2021)*, we used forceps to repeatedly stimulate an adult *Ph. occipitalis jessoensis*. This treatment caused the beetle to release all the chemicals. Each beetle repeatedly bombed before exhausting its chemicals. We then used the same procedure as for the control beetles to observe whether a bullfrog successfully attacked the treated beetle in a transparent plastic case (length × width × height, 120 × 85 × 130 mm). In total, 27 bullfrogs and 27 beetles were used in the experiments. The sample size was determined based on the previous study (*Sugiura, 2018*).

All experiments were undertaken in accordance with the Kobe University Animal Experimentation Regulations (Kobe University's Animal Care and Use Committee, 30–01). No bullfrogs were injured during the feeding experiments. Because the release of bullfrogs into the wild is banned in Japan, the bullfrogs used in this study were euthanized by $CO_2$ asphyxiation after all experiments were completed.

## Data analysis

Fisher's exact test was used to compare the rejection rate of control beetles with that of treated beetles by bullfrogs; the rejection rate of bombardier beetles collected from invaded sites with that of beetles from non-invaded sites; and the rejection rate of control beetles by bullfrogs with that by native pond frogs. Odds ratios and 95% confidence intervals (CIs) were also calculated. Data from *Sugiura (2018)* were also used as the rejection rate by the native pond frog *Pe. nigromaculatus*. Welch's *t*-test was used to compare the body size of bullfrogs and bombardier beetles between control and treated experiments. A generalized linear model (GLM) with a binomial error distribution and logit link was used to determine the factors contributing to the rejection of bombardier beetles by bullfrogs (cf. *Sugiura, 2018*). The rejection (1) or predation (0) of bombardier beetles by bullfrogs was used as the response variable. Beetle weight, bullfrog weight, the beetle weight × bullfrog weight interaction, and beetle treatment (control or treated) were included as fixed factors. A quasi-binomial error distribution was used rather than a binomial error distribution, which is necessary if the residual deviance is smaller (underdispersion) or larger (overdispersion) than the residual degrees of freedom (cf. *Sugiura, 2018*). All the tests were

performed at the 0.05 significance level. All analyses were performed using R ver. 3.5.2 (*R Core Team, 2018*).

## RESULTS

All bullfrogs (*n* = 27) opened their mouths to attack bombardier beetles (control beetles); however, 26 bullfrogs (96.3%) rejected bombardier beetles (Table 1). The rejection was not due to satiation because 25 (96.2%) of the bullfrogs that rejected beetles ate mealworms immediately after the rejection. Only one bullfrog (3.7%) successfully swallowed and digested the beetle (Table 1). The swallowed beetle did not bomb when attacked. This beetle was relatively old (its sampling date was the earliest among all beetles). Two bullfrogs (7.4%) rejected the beetles before being bombed; one bullfrog (3.7%) stopped attacking the beetle immediately after its tongue touched it, and one bullfrog (3.7%) spat out the beetle <1 s after taking it into its mouth (Table 1). Two bullfrogs (7.4%) were bombed before taking the beetles into their mouths and immediately stopped the attack (Table 1). Twenty-two bullfrogs (81.5%) were bombed within 5 s of taking the beetles into their mouths and then spat them out within 2 s after being bombed (Video S2; Fig. 2; Table 1). The collection sites of bombardier beetles did not influence the rejection rates by bullfrogs (Fisher's exact test, *P* = 1.0, odds ratio [95% CI] = ∞ [0.01283594–∞]); 94.4% of the beetles (*n* = 18) collected from bullfrog-invaded sites and 100% of the beetles (*n* = 9) from non-invaded sites were rejected by bullfrogs. The behavioral responses of bullfrogs to bombardier beetles were compared with those of the native pond frog species *Pe. nigromaculatus* (Fig. 3). The rate of swallowing and rejection of beetles did not significantly differ between the two species (Fisher's exact test, *P* = 1.0, odds ratio [95% CI] = 0.506127 [0.008172–10.284953]), but the rate of rejection before bombing significantly differed between the two species (Fig. 3; *P* = 0.000005, odds ratio [95% CI] = 0.042546 [0.004042–0.229305]).

When treated beetles that were unable to bomb were provided, all bullfrogs (*n* = 27) attacked the beetles. Twenty-one bullfrogs (77.8%) successfully swallowed and digested treated beetles, while six bullfrogs (22.2%) spat out treated beetles within 7 s of taking them into their mouths (Table 1). All of the bullfrogs that rejected treated beetles (*n* = 6) ate mealworms after the rejection. The rejection rate of treated beetles by bullfrogs (22.2%) differed significantly from that of control beetles (96.3%; Table 1; Fisher's exact test, *P* = 0.00000002, odds ratio [95% CI] = 0.01246560 [0.00026351–0.10342453]).

The body lengths and weights of treated beetles were not significantly different from those of control beetles (Table 2; t-test, *P* = 0.16–0.86). The snout–vent lengths and weights of bullfrogs that attacked control beetles were not significantly different from those of bullfrogs that attacked treated beetles (Table 2; t-test, *P* = 0.50–0.66). The GLM results indicated that the rejection rate of bombardier beetles by bullfrogs was influenced by beetle treatment, but not by the body size of either bombardier beetles or bullfrogs (Table 3).

## DISCUSSION

The American bullfrog *L. catesbeianus* can eat any animals smaller than itself (*Adriaens, Devisscher & Louette, 2013*). Consequently, introduced bullfrogs have negatively affected

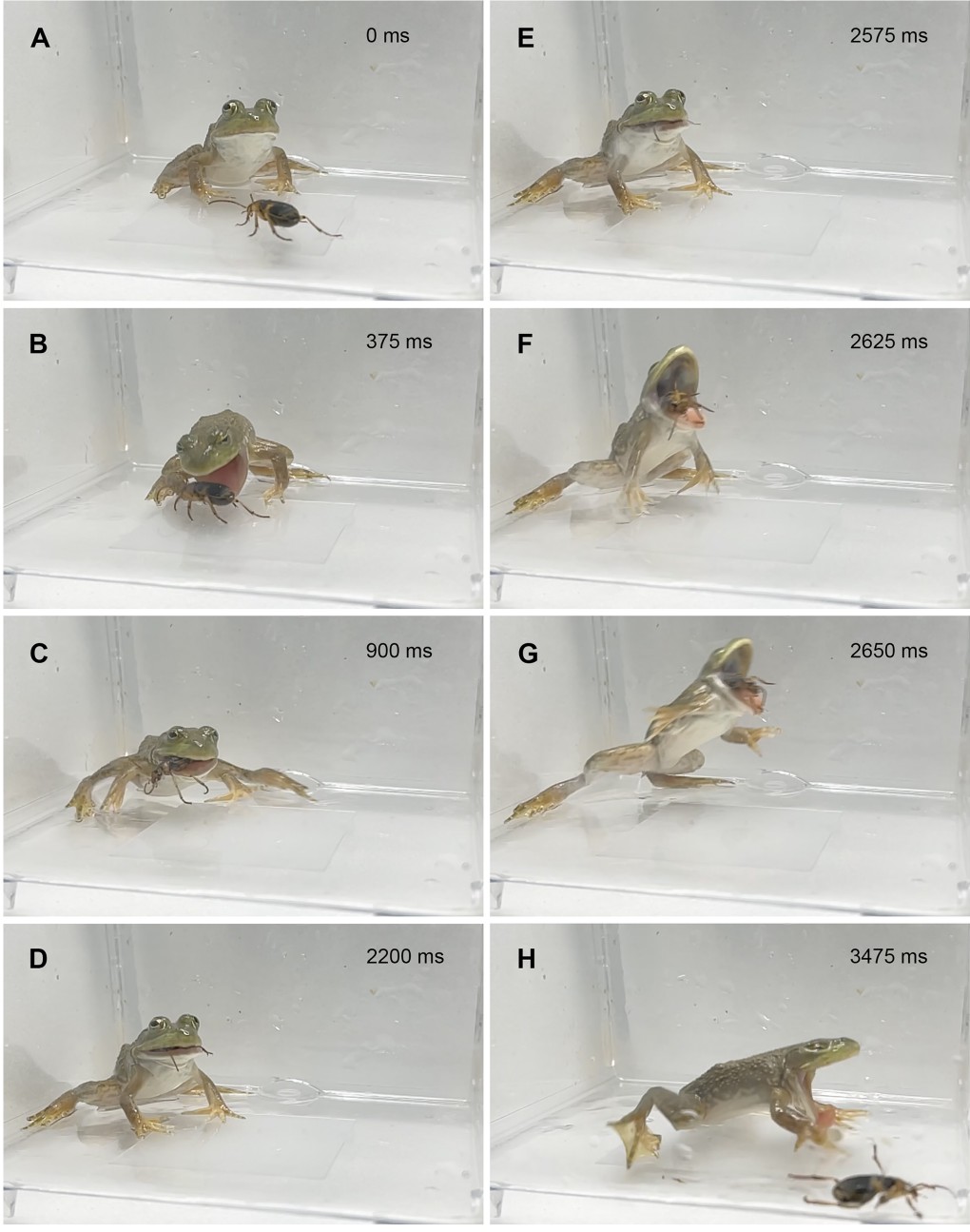

**Figure 2 Temporal sequence of the bullfrog *Lithobates catesbeianus* rejecting a control adult *Pheropsophus occipitalis jessoensis*.** (A) 0 ms. (B) 375 ms. (C) 900 ms. (D) 2,200 ms. (E) 2,575 ms. (F) 2,625 ms. (G) 2,650 ms. (H) 3,475 ms. The bullfrog spat out the beetle after taking it into its mouth. Bombing by the beetle was audible and the ejected vapor (E) was observed just before the bullfrog spat out the beetle (see Video S2). Credit: Shinji Sugiura and Tomoki Date.

native arthropods and amphibians through direct predation in invaded areas (*Kats & Ferrer, 2003*; *Li et al., 2011*; *Adriaens, Devisscher & Louette, 2013*; *Gobel, Laufer & Cortizas, 2019*; *Groffen et al., 2019*; *Nakamura & Tominaga, 2021*). Although the native bombardier beetle *Ph. occipitalis jessoensis* has reportedly been identified in the stomach contents of

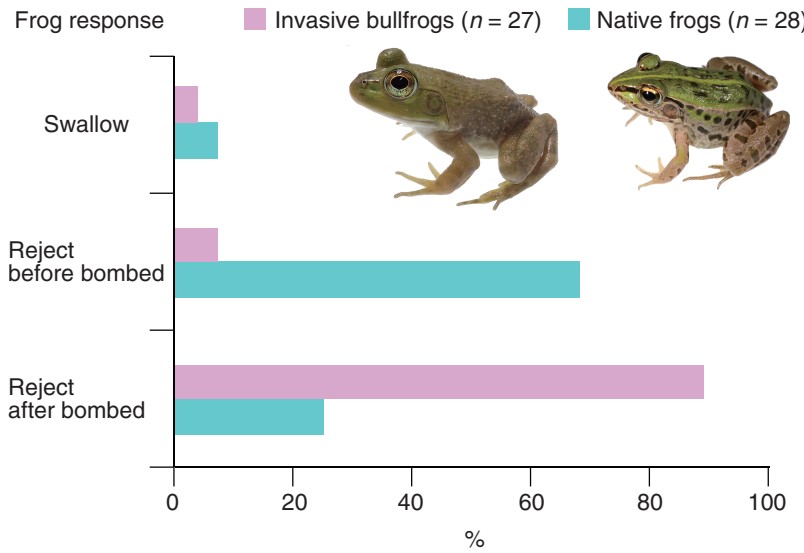

Frog response — Invasive bullfrogs (*n* = 27) — Native frogs (*n* = 28)

Swallow

Reject before bombed

Reject after bombed

%

**Figure 3 Behavioral responses of the invasive bullfrog *Lithobates catesbeianus* and the native pond frog *Pelophylax nigromaculatus* to adults of the bombardier beetle *Pheropsophus occipitalis jessoensis*.** Swallow: bullfrogs or frogs successfully swallowed control beetles. Reject before bombed: bullfrogs or frogs stopped attacking control beetles before being bombed. Reject after bombed: bullfrogs or frogs rejected control beetles after being bombed. The graph showing data for *Pe. nigromaculatus* was taken from *Sugiura (2018)*. Photo credit: Shinji Sugiura.

**Table 1 Responses of the bullfrog *Lithobates catesbeianus* to control and treated adults of the bombardier beetle *Pheropsophus occipitalis jessoensis*.**

| Frog response[a] | Frog behavior[b] | Control beetles[c] % (*n*) | Treated beetles[c] % (*n*) |
|---|---|---|---|
| Eat | Swallow | 3.7 (1) | 77.8 (21) |
| Reject (subtotal) | | 96.3 (26) | 22.2 (6) |
| Reject before bombed | Stop attack | 3.7 (1) | 0.0 (0) |
| | Spit out | 3.7 (1) | 22.2 (6) |
| Reject after bombed | Stop attack | 7.4 (2) | – |
| | Spit out | 81.5 (22) | – |
| Total | | 100.0 (27) | 100.0 (27) |

Notes:
[a] Eat: bullfrogs successfully ate beetles. Reject before bombed: bullfrogs rejected beetles before or without being bombed. Reject after bombed: bullfrogs rejected beetles after being bombed.
[b] Swallow: bullfrogs successfully swallowed beetles. Stop attack: bullfrogs stopped attacking beetles before taking them into their mouths. Spit out: bullfrogs spat out beetles after taking them into their mouths.
[c] Control beetles and treated beetles are the *Pheropsophus occipitalis jessoensis* that were able and unable to discharge hot chemicals, respectively.

introduced bullfrogs in Japan (*Mori, 2008*; *Matsumoto, Suwabe & Karube, 2020*), our laboratory experiments showed that almost all bullfrogs rejected *Ph. occipitalis jessoensis* before swallowing them. Therefore, *Ph. occipitalis jessoensis* can successfully repel invasive bullfrogs using a chemical weapon. To our knowledge, this is the first study to demonstrate the successful defense of a native insect species against invasive bullfrogs. However, this study may reflect limited aspects of prey–predator interactions between native bombardier beetles and invasive bullfrogs, as it was not designed to assess the potential effects of

**Table 2 Sizes of the bombardier beetle *Pheropsophus occipitalis jessoensis* and the bullfrog *Lithobates catesbeianus* used in this study.**

| Species | Boy size | Treatment | | Statistical comparison | |
|---|---|---|---|---|---|
| | | Control beetles $n = 27$ | Treated beetle $n = 27$ | t value | P value |
| Bombardier beetle | Body length (mm)[a] | 17.6 ± 0.2 | 17.6 ± 0.2 | 0.25 | 0.80 |
| | | (15.2–20.2) | (15.5–19.6) | | |
| | Boy weight (mg)[a] | 265.8 ± 12.4 | 241.7 ± 11.4 | 1.43 | 0.16 |
| | | (149.1–411.3) | (146.5–376.2) | | |
| | | | 268.9 ± 12.4 | (−0.18)[c] | (0.86)[c] |
| | | | (164.4–409.9)[b] | | |
| Bullfrog | Snout–vent length (mm)[a] | 48.2 ± 0.8 | 47.8 ± 0.7 | 0.44 | 0.66 |
| | | (43.5–59.6) | (42.2–57.3) | | |
| | Body weight (mg)[a] | 9206.6 ± 554.7 | 8720.9 ± 458.1 | 0.68 | 0.50 |
| | | (6136.9–18257.1) | (5575.6–16763.8) | | |

Notes:
[a] Values are the mean ± standard error (range: minimum–maximum).
[b] Body weight of bombardier beetles before treatment.
[c] Statistical result of a comparison between treated beetles (before treatment) and control beetles.

**Table 3 Results of a generalized linear model (GLM) identifying factors affecting whether the bullfrog *Lithobates catesbeianus* rejected the bombardier beetle *Pheropsophus occipitalis jessoensis*.**

| Response variable | Explanatory variable (fixed effect) | Coefficient estimate | SE | t value | P value |
|---|---|---|---|---|---|
| Rejection[a] | Intercept | 11.68 | 7.466 | 1.564 | 0.12418 |
| | Beetle treatment[b] | −5.389 | 1.673 | −3.222 | 0.00226 |
| | Beetle size (weight) | −0.02395 | 0.0308 | −0.777 | 0.44061 |
| | Frog size (weight) | −0.0007511 | 0.000699 | −1.075 | 0.28783 |
| | Beetle size × frog size | 0.000002211 | 0.000003243 | 0.682 | 0.49872 |

Notes:
[a] A quasi-binomial error distribution (rather than a binomial error distribution) was used because the residual deviance was smaller than the residual degrees of freedom.
[b] Control beetles were used as a reference.

bullfrog size and learning on successful defenses of *Ph. occipitalis jessoensis*. Considering this limitation, we discuss the importance of bombing behavior as a pre-existing defense of *Ph. occipitalis jessoensis* against invasive bullfrogs, and the potential impact of invasive bullfrogs on native *Ph. occipitalis jessoensis*.

Some native species can evolve a tolerance to or defense against invasive predators (*Strauss, Lau & Carroll, 2006*). However, all adults of *Ph. occipitalis jessoensis* collected from non-bullfrog-invaded sites could successfully defend against bullfrogs, suggesting that the pre-existing defense of *Ph. occipitalis jessoensis* was strong enough to repel bullfrogs. Like invasive bullfrogs, the native pond frog *Pe. nigromaculatus* has been shown to frequently reject *Ph. occipitalis jessoensis* under laboratory conditions (*Sugiura, 2018*). Because both the native frog and the introduced bullfrog are frequently found in the same habitats in Japan (*Kambayashi et al., 2016*; *Sato, 2016*; *Tawa & Sagawa, 2017*), the defenses of *Ph. occipitalis jessoensis* that originally functioned against native frogs could play an important role in repelling invasive bullfrogs.

*Sugiura (2018)* showed that 67.9% of the native frog *Pe. nigromaculatus* rejected *Ph. occipitalis jessoensis* before being bombed. When dead adults of *Ph. occipitalis jessoensis* were provided, 71.4% of *Pe. nigromaculatus* rejected them (*Sugiura, 2018*). The native frog species stopped attacking live and dead *Ph. occipitalis jessoensis* immediately after their tongues contacted them, indicating that this frog species may avoid being bombed by detecting chemicals on the surface of the beetle (*Sugiura, 2018*). The present study showed that only 7.4% of bullfrogs rejected *Ph. occipitalis jessoensis* before being bombed. Therefore, bombing by *Ph. occipitalis jessoensis* is much more important for a successful defense against invasive bullfrogs than against native frogs. Unlike native frogs, bullfrogs may not use their tongue to detect a deterrent chemical or the physical characteristics of *Ph. occipitalis jessoensis*.

Adults of *Ph. occipitalis jessoensis* were found in the stomach contents of introduced bullfrogs in Japan (*Mori, 2008*; *Matsumoto, Suwabe & Karube, 2020*), although our results showed that almost all bullfrogs failed to eat adult *Ph. occipitalis jessoensis*. At least three factors may help to explain this inconsistency: high encounter rates between adult *Ph. occipitalis jessoensis* and bullfrogs; deficiency of defensive chemicals in old adult *Ph. occipitalis jessoensis*; and different body sizes of the bullfrogs used in the present and previous studies. First, because *Ph. occipitalis jessoensis* is commonly found in grassland and farmland around ponds, lakes, and paddy fields invaded by bullfrogs, adults of *Ph. occipitalis jessoensis* frequently encounter bullfrogs in Japan. The high encounter rate between *Ph. occipitalis jessoensis* and bullfrogs could result in successful predation events by bullfrogs even when the overall success rate of predation on *Ph. occipitalis jessoensis* by bullfrogs is low. To the second point, old adults of *Ph. occipitalis jessoensis* that are unable to produce enough defensive chemicals can easily be eaten by bullfrogs. In our experiment, the swallowed adult *Ph. occipitalis jessoensis* was relatively older than the other beetles and did not bomb when attacked. Lastly, *Matsumoto, Suwabe & Karube (2020)* found an adult *Ph. occipitalis jessoensis* in the stomach content of a juvenile bullfrog (snout–vent length: 83 mm) that was larger than the juveniles used in our experiments (snout–vent length: 43.4–59.6 mm). This suggests that *Ph. occipitalis jessoensis* may fail to defend itself against bullfrog adults and large juveniles. The importance of predator size for the successful defense of *Ph. occipitalis jessoensis* was suggested by *Sugiura & Sato (2018)* who showed that adult and large juvenile toads could more frequently eat adult *Ph. occipitalis jessoensis* than the small juveniles. However, juvenile bullfrogs of the size used in this study are much more abundant than the adults and large juveniles in invaded areas in Japan (*Sato & Nishihara, 2017*; *Matsumoto, Suwabe & Karube, 2020*). Therefore, unlike other native insect species, the native bombardier beetle *Ph. occipitalis jessoensis* may not suffer predation by invasive bullfrogs. However, no studies have quantitatively compared the abundance of *Ph. occipitalis jessoensis* between bullfrog-invaded and non-invaded areas. Further studies are needed to demonstrate the impacts of invasive bullfrogs on *Ph. occipitalis jessoensis*.

## CONCLUSIONS

Some native animal species can tolerate invasive predators by evolving defenses against the predators (*Vermeij, 1982*; *Strauss, Lau & Carroll, 2006*) or using pre-existing defensive strategies (*Davis, Epp & Gabor, 2012*; *Carthey & Banks, 2014*). Although bombardier beetles possess chemical weapons to deter various types of predators (*Eisner, Eisner & Siegler, 2005*; *Sugiura, 2020a*), how they defend against invasive predators has been unclear. Our laboratory experiments demonstrated that the native bombardier beetle *Ph. occipitalis jessoensis* was able to repel invasive bullfrogs by bombing. Because *Ph. occipitalis jessoensis* can defend itself against the native pond frog *Pe. nigromaculatus* (*Sugiura, 2018*) and other native predators (*Sugiura & Sato, 2018*; *Kojima & Yamamoto, 2020*; *Sugiura, 2021*) in Japan, *Ph. occipitalis jessoensis* uses its pre-existing defense to defend against invasive bullfrogs, which occupy a similar niche to that of native pond frogs.

## ACKNOWLEDGEMENTS

We are grateful to M. Hayashi for helping to collect bombardier beetles.

### Funding

The following grant information was disclosed by the authors: Grant-in-Aid for Scientific Research (JSPS KAKENHI): 19K06073. The funders had no role in study design, data collection and analysis, decision to publish, or preparation of the manuscript.

### Grant Disclosures

The following grant information was disclosed by the authors:
Grant-in-Aid for Scientific Research (JSPS KAKENHI): 19K06073.

### Competing Interests

The authors declare that they have no competing interests.

### Author Contributions

- Shinji Sugiura conceived and designed the experiments, performed the experiments, analyzed the data, prepared figures and/or tables, authored or reviewed drafts of the article, and approved the final draft.
- Tomoki Date performed the experiments, prepared figures and/or tables, and approved the final draft.

### Animal Ethics

The following information was supplied relating to ethical approvals (*i.e.*, approving body and any reference numbers):

The authors followed Kobe University's Animal Experimentation Regulations and obtained approval from Kobe University's Animal Care and Use Committee (30–01).

## Data Availability

The raw data are available at figshare: Sugiura, Shinji; Date, Tomoki (2022): Data from: Bombardier beetles repel invasive bullfrogs. figshare. Dataset. https://doi.org/10.6084/m9.figshare.19439453.

## Supplemental Information

Supplemental information for this article can be found online at http://dx.doi.org/10.7717/peerj.13805#supplemental-information.

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
