# Peer review of "Bombardier beetles repel invasive bullfrogs"

_PeerJ, doi:10.7717/peerj.13805_

## Round 0.1 · original submission · Major Revisions

Please revise your manuscript to address the reviewers' concerns.

Reviewer 1 ·

Basic reporting

The manuscript entitled "Bombardier beetles repel invasive bullfrogs", by Sugiura and Date, provides information on the relationship (under laboratory conditions) occurring between an invasive species, The American bullfrog Lithobates catesbeianus and a native beetle, Pheropsophus (Stenaptinus) occipitalis jessoensis, referred to as prey in previous studies.
Interactions were evaluated from a behavioral perspective, considering several variables, such as the size of both prey and predator and the ability to discharge chemicals as a defensive strategy of the prey.
The study appears to be one of the first in the field considering the potential effect of an invasive species on native fauna.
However, some important aspects should be focused, such as the ecological importance of carabid beetles, known as bioindicators (Rainio, J., & Niemelä, J. (2003). Ground beetles (Coleoptera: Carabidae) as bioindicators. Biodiversity & Conservation, 12(3), 487-506.) and for providing ecosystem services (Akhil, S. V., & Thomas, S. K. (2018). Bombardier beetles (Coleoptera: Carabidae: Brachininae) of India–notes on habit, taxonomy and use as natural bio-control agents. Frontiers in biological research, 1-25).
Moreover also the secretion seem to represent an evolutionary conserved and species-specific aspect among carabids (Giglio, A., Vommaro, M. L., Brandmayr, P., & Talarico, F. (2021). Pygidial Glands in Carabidae, an Overview of Morphology and Chemical Secretion. Life, 11(6), 562) to be considered by a predator-prey coevolution point of view. This aspect should be explained in more detail in both the introduction and the discussion.

Experimental design

EXPERIMENTAL DESIGN
The experimental design is clear and of easy comprehension and the methods are described with sufficient details.
However, to better visualize the sample sites, a map could be provided as a supplemental figure, highlighting all collection sites.
L. 158: It is unclear why different sample sizes were used from the invaded and non-invaded sites, was the abundance of beetles different?

Validity of the findings

VALIDITY OF RESULTS
The results are interesting from an ecological point of view, however I would suggest considering the following point.
1. I would suggest including the relationship between body condition(size and weight) and predatory activity (rejection, swallowing) in the analyses using a GLM, as previously mentioned in Sugiura, 2018, in order to statistically assess the interaction between these variables.
2. The authors argue the difference between native and invasive frog in beetle rejection by considering the ability of native frogs to detect the presence of chemicals on the cuticle, given a co-evolution prey-predator strategy. Was the response tested in the native frogs by exposing them to the treated (beetles not able to bomb) and control beetle? To support the argument, even beetles that do not discharge chemicals should be repelled by frogs.
3. It is unclear if a difference in species abundance has been found at sites where invasive bullfrog and non-invasive sites have been reported. Please consider previous data if available or highlight the lack of information if this is not the case.
4. In my opinion the concluding section is not focused on the main findings of the manuscript, please reformulate the section indicating the main findings and addressing the goals stated in the introduction and add research perspectives.

Additional comments

I would also like to report some minor changes and comments divided by section:

INTRODUCTION
The introduction provides a comprehensive overview of the topic, but it is repetitive in a few sections (L. 93-94 and L. 108). Please delete the repetitive paragraph to make it a more fluent read.

MATERIAL AND METHODS
1. The paragraph should be dedicated to the description of experimental design, protocols, and techniques applied. All background information should be given in the introduction, L. 127-136 sentences seem to be irrelevant in the context of the paragraph, please remove them or move them to the introductory section.
2. L. 158 "Thirty-nine and 15 beetles were collected “ Please, standardize writing of numbers (as word or numeric characters).
3. The sentence at L.143 is repeated at L. 160, please delete one.

RESULTS
The results section should be related to data analysis, without assumptions, move the sentence at L. 232-234 to the discussion.

DISCUSSION
1. Authors should comment on all figures and tables in the results section, please remove all references to figures and tables in the discussion (e.g., L. 248; 258; 260; 264; 268).
2. The discussion provides interesting arguments, but sounds speculative in some statement, please rephrase or remove L. 253-254

Reviewer 2 ·

Basic reporting

General Assessment:
The authors have presented a very well-written and executed study on the efficacy of the "bombing" defense of Pheropsophus occipitalis jessoensis against the invasive Lithobates catesbeianus in Japan. The data and their analysis strongly support the basic conclusion that this defense is very effective. My main criticisms relate to the conclusions drawn from this observation which need to be re-considered and/or expanded on.

A. Clear and unambiguous, professional English used throughout
Yes, this article was written very well.

B. Literature references, sufficient field background/context provided.
There is a problem with the use of "co-evolution" in the introduction (and discussion). "Co-evolution" has a very specific meaning in the literature, involving reciprocal evolutionary change in two or more lineages. This term does not necessarily apply to all predator-prey interactions. It seems the authors' use of the term does not meet the usual, strict definition. It would be more appropriate to say something along the lines of "...do not share a history of predator-prey interaction..." and remove references to "co-evolution" in this case.

Line 122-123, 302-303: More context is required here. Was it 1 individual beetle that was found? Was it many beetles eaten by a few frogs? A few beetles eaten by many frogs? Many beetles eaten by many frogs?

C. Professional article structure, figures, tables. Raw data shared.
Yes, these were well done.

D. Self-contained with relevant results to hypotheses.
This is borderline in my opinion. While this study does represent a logical unit of publication, it is very similar to previously published works by the authors (Sugiura & Takuya 2018 [10.1098/rsbl.2017.0647] and Sugiura 2018[10.7717/peerj.5942]), which mainly differ only in the frog species they test.

Experimental design

A. Original primary research within Aims and Scope of the journal.
Yes.

B. Research question well defined, relevant & meaningful. It is stated how research fills an identified knowledge gap
Yes.

C. Rigorous investigation performed to a high technical & ethical standard.
Potential effects on the results from startle and learning are not addressed with this experimental design. If frogs had been tested multiple times over multiple days, they may have overcome this defense. This limits the generalization of these results to wild interactions and could also contribute to the observation of these beetles in wild Lithobates catesbeianus diets.

D. Methods described with sufficient detail & information to replicate.
Sex of frogs & beetles is not reported. Perhaps this does not matter much in this case, but maybe the authors could address whether they mainly used one sex or another in these experiments.

Validity of the findings

A. All underlying data have been provided; they are robust, statistically sound, & controlled.
Line 218 (& Results): Please provide a measure of effect size in your reporting of statistical results alongside the p-values. Odds ratios should be a good choice for Fisher's Exact Test.

B. Conclusion are well stated, linked to original research question & limited to supporting results.
These frogs may have not have evolved with Ph. occipitalis jessoensis as natural prey, but they likely have evolved in habitats containing other species of bombardier beetles. So, bombing is not expected to be an "exotic" defense from the perspective of these frogs. Can the authors discuss the implication of this further in the context of the setup for this study (e.g., Lines 295-301)

If failure of defense is ~4%, this might not be surprising that they are sometimes found in their diets, especially if encounter rates are high. Is there any estimate of the encounter rates between these species? With a sufficiently high number of encounters, even a low percentage of eaten beetles could translate to discovering these insects in the diets of these frogs.

Line 309-310: Beetles sometimes fail to discharge after multiple discharges. Some information about bombing behavior as beetles age would be useful too (if known). The efficacy of the defense might wane over time and explain higher average predation rates in the wild vs lab. In addition, knowing how frequently wild beetles are be able to perform their bombing behavior would be useful. How long does it take to regenerate the bombing defense once exhausted? Could the beetles observed in the diets of these frogs be attributable to old or "exhausted" beetles that could not perform their bombing effectively? These topics and potential alternative explanations should be addressed further in the Discussion section.

---

## Round 0.2 · accepted · Accept

Thank you for your careful revisions to the manuscript.

Reviewer 1 ·

Basic reporting

The authors have adequately addressed all comments and criticisms, revising the manuscript according to suggestions. I suggest accepting the manuscript without further requests.

Experimental design

The clarifications and modifications, such as the addition of Fig. S1, have made the experimental design more understandable.
The only note I would like to make is that it is possible to sex adult carabid beetles by observing the fore legs; the tarsi only in males are covered ventrally by bristles and brush. I would suggest that this detail be taken into account for further study, also because there may be a different response between males and females.

Validity of the findings

GLM analysis enriched the manuscript with clearer evidence on the role of treatments.